# Distance Map Loss Penalty Term for Semantic Segmentation

**Francesco Calivá**[*1]    Francesco.Caliva@ucsf.edu
**Claudia Iriondo**[*1,2]    Claudia.Iriondo@ucsf.edu
**Alejandro Morales Martinez**[1,2]    Alejandro.MoralesMartinez@ucsf.edu
**Sharmila Majumdar**[1]    Sharmila.Majumdar@ucsf.edu
**Valentina Pedoia**[1]    Valentina.Pedoia@ucsf.edu

[*]*Both authors contributed equally.*

[1]*Department of Radiology and Biomedical Imaging, University of California, San Francisco*

[2]*Department of Bioengineering, University of California, Berkeley*

## Abstract

Convolutional neural networks for semantic segmentation suffer from low performance at object boundaries. In medical imaging, accurate representation of tissue surfaces and volumes is important for tracking of disease biomarkers such as tissue morphology and shape features. In this work, we propose a novel distance map derived loss penalty term for semantic segmentation. We propose to use distance maps, derived from ground truth masks, to create a penalty term, guiding the network's focus towards hard-to-segment boundary regions. We investigate the effects of this penalizing factor against cross-entropy, Dice, and focal loss, among others, evaluating performance on a 3D MRI bone segmentation task from the publicly available Osteoarthritis Initiative dataset. We observe a significant improvement in the quality of segmentation, with better shape preservation at bone boundaries and areas affected by partial volume. We ultimately aim to use our loss penalty term to improve the extraction of shape biomarkers and derive metrics to quantitatively evaluate the preservation of shape.

**Keywords:** penalized loss, bone segmentation, distance maps, Osteoarthritis Initiative, magnetic resonance imaging, 3D convolutional neural networks

## 1. Introduction

The segmentation of medical images enables the quantitative analysis of anatomical structures. In both 2D and 3D medical imaging data, state of the art segmentation performance has been achieved using Convolutional Neural Networks, with U-Net (Ronneberger et al., 2015), V-Net (Milletari et al., 2016), and variants thereof. In this work, the original V-Net architecture was chosen as the end-to-end encoder-decoder architecture, because of its peculiar capability of learning a residual function within each down- and up- sampling stage. This alleviates the problem of overfitting and vanishing gradients, with the added benefit of faster convergence (He et al., 2016). The loss function proposed in (Milletari et al., 2016), is the baseline of our experiments. V-Net aims to minimize the soft Dice loss, derived from the Dice coefficient

$$Dice = \frac{2\sum_i^N p_i g_i}{\sum_i^N p_i^2 + g_i^2} \tag{1}$$

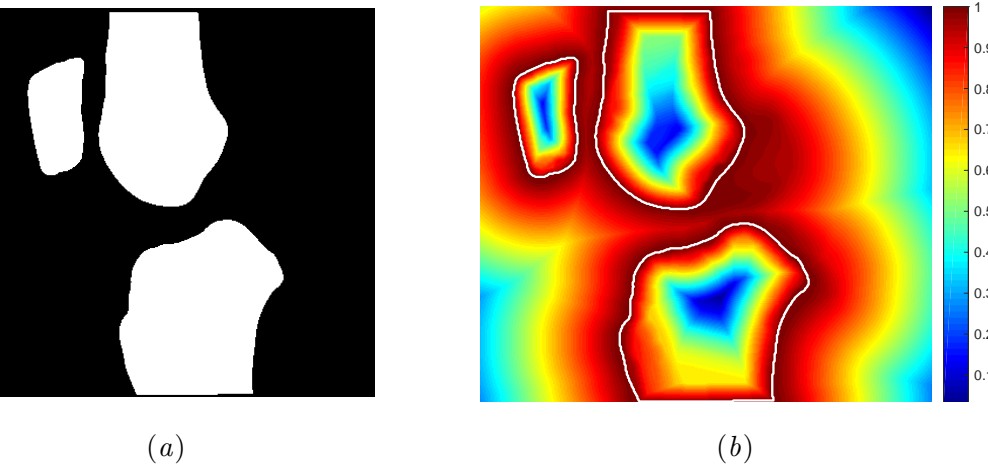

$$(a) \qquad\qquad\qquad (b)$$

Figure 1: **(a)** Ground truth segmentation and **(b)** distance map, bone boundaries in white

where the sum runs over all the $p \in P$ and $g \in G$ volume voxels of the generated segmentation and the relative ground truth masks respectively.

We conducted an initial experiment using a V-Net architecture to segment knee bones in 3D MRIs. In agreement with (Milletari et al., 2016) we observed superior segmentation performance when using Dice loss compared to the weighted log-likelihood loss. Irrespective of choice of loss function, most errors were located at the proximity of bone boundaries. This work proposes a simple strategy to penalize segmentation errors at object boundaries utilizing distance maps generated on the segmentation ground truth. The approach is similar to (Kervadec et al., 2018). Nevertheless, we train with a distance based loss penalty from the beginning, while (Kervadec et al., 2018) proposes a fine tuning like strategy. Furthermore, we extend the approach to a 3D and multi-class context, and we are driven by different motivations: in (Kervadec et al., 2018), the goal is to deal with highly imbalanced dataset, whereas our focus is accurate segmentation of object boundaries. We also conduct a more thorough comparison with other state of the art attention-based losses. Finally, application of our method to highly imbalanced datasets is straightforward.

## 2. Methods and Experiments

The Osteoarthritis Initiative (OAI) dataset is comprised of knee MR scans from 4,796 unique patients scanned at 10 different time points, MR acquisition described in (Norman et al., 2018).Forty unique patients were manually segmented obtaining ground truth masks for the distal femur, proximal tibia, and patella. These were used to evaluate our proposed method with a 25/5/10 train/valid/test split. Error-penalizing distance maps (Figure 1) were generated by computing the distance transform on the segmentation masks and then reverting them, by voxel-wise subtracting the binary segmentation from the mask overall max distance value. This procedure aims to compute a distance mask where pixels in proximity of the bones are weighted more, compared to those located far away. An identical

procedure was conducted on the negative version of the segmentation mask to calculate a distance map inside the bones. To account for differences in bone size, with the femur being 1.6 and 16 times larger than tibia and patella respectively, inner distance maps for each bone were independently computed and subsequently combined. The generated maps $\Phi$ were utilized to penalize prediction errors during training. In practice, the aim is to minimize the "penalized" multi-class cross entropy loss $\mathscr{L}$ in Equation (2),

$$\mathscr{L} = \frac{1}{N} \sum_{i=1}^{N} (1 + \Phi) \odot \sum_{j=1}^{K} -y_j \log \hat{y}_j \tag{2}$$

where the two sums run over the $i$ samples and the $j$ classes, and $\odot$ is the Hadamard product. Adding 1 to $\Phi$ has the effect of mitigating the vanishing gradient issue. We benchmarked the proposed penalizing term against commonly used loss functions, including soft-dice loss, focal loss (Lin et al., 2017), and the confident predictions penalizing loss proposed in (Pereyra et al., 2017). A V-Net architecture was trained using mini-batch Gradient Descent with Adam Optimizer (Kingma and Ba, 2014) (learning rate $10^{-4}$) and random in-plane rotations as augmentations. MATLAB (Matlab, 1760) and Tensorflow 1.12 (Abadi et al., 2016) were run on an Intel®Xeon (R) Gold 6130 CPU @ 2.10GHz, four GPUs and 376GB of RAM.

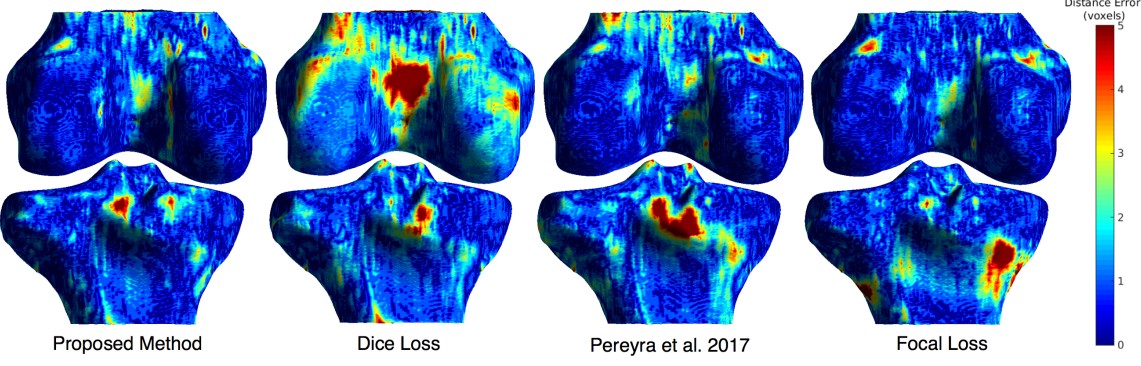

Figure 2: Posterior view of distal femur and proximal tibia for a single test patient, predicted segmentation's absolute distance from ground truth. 0 is a perfect segmentation.

## 3. Results and Conclusions

Predicted segmentation masks were post-processed by applying 3D morphological closing and extraction of the three largest connected components. To demonstrate the utility of our loss penalty term, we compare it to other successful methods in Figure 2 and Figure 3, using error maps and the following metrics: global Dice score (G-DSC), boundary Dice score (B-DSC), and its relaxed version which expands boundaries by a certain tolerance. Our method produces high-quality segmentations, with accurate results even in regions with significant partial voluming (intercondyle notch, tibial condyles). B-DSC of our proposed loss shows

a significant improvement in edge detection ($28.83 \pm 4.45\%$ vs Dice loss $26.73 \pm 5.40\%$ vs (Pereyra et al., 2017) $25.81 \pm 3.02\%$ vs focal loss $26.70 \pm 4.27\%$). This superior performance is maintained globally (G-DSC) ($96.42 \pm 0.80\%$ vs Dice loss $96.34 \pm 1.21\%$ vs (Pereyra et al., 2017) $95.96 \pm 1.30\%$ vs focal loss $95.00 \pm 1.00\%$). We observed that guiding the network with a shape-aware loss function is a promising method to improve segmentation performance.

## Acknowledgments

This work was supported by the NIH/NIAMMS R00AR070902

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

## Appendix A. Additional Results

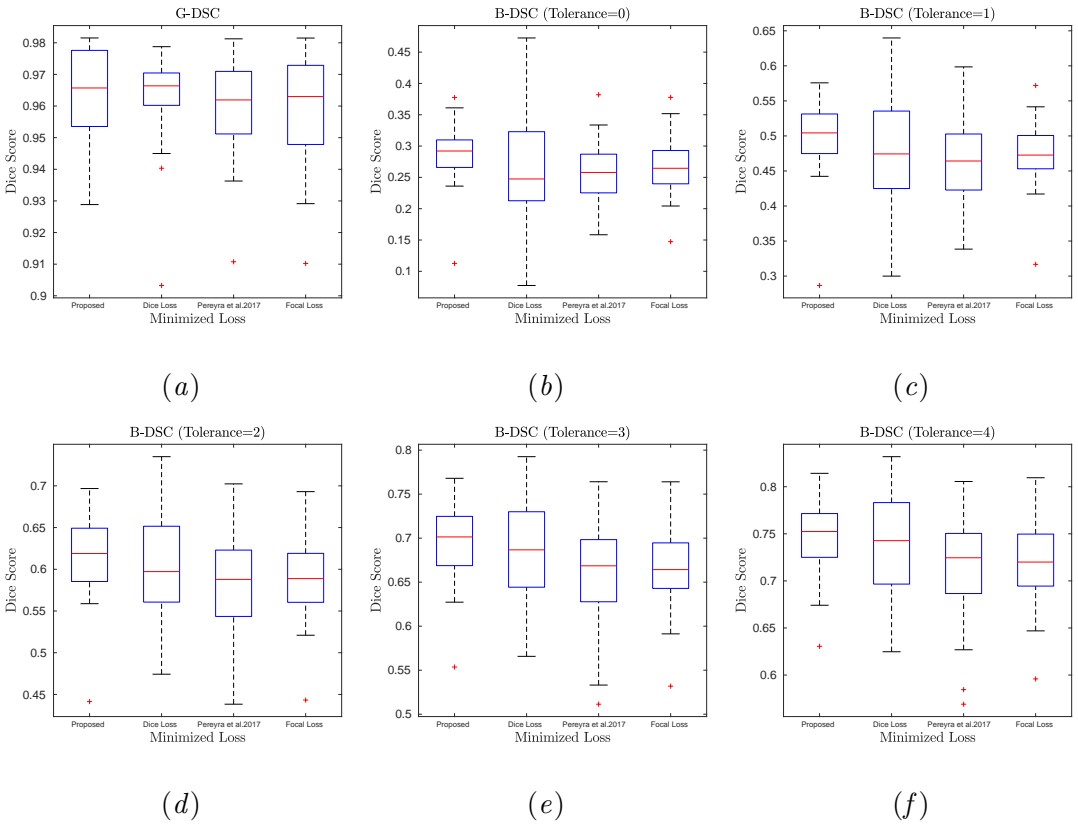

$(a)$        $(b)$        $(c)$

$(d)$        $(e)$        $(f)$

Figure 3: Performance comparison of the proposed distance map penalizing loss term against the Dice Loss function, confident predictions penalizing loss and the focal loss. **(a)** Global Dice Score Coefficient G-DSC, **(b)** Boundary Dice Score Coefficient B-DSC, and **(c-f)** relaxed B-DSC tolerance 1 to 4 voxels are reported.

