# OpenReview forum: "Distance Map Loss Penalty Term for Semantic Segmentation"
_MIDL.io/2019/Conference/Abstract — MIDL Abstract 2019_

### Official Review · AnonReviewer2 · 2019-04-30
**A penalty for segmentation**

**Rating:** 3
**Confidence:** 2

**Review:**

The paper presents a penalty term based on the distance map for penalizing the segmentation errors during training. V-net is used to segment know bones in 3d MRI. The experimental results are promising with the penalty term improving the results.

Suggestion:
The work is well written and is also a bit similar to the MIDL 2019 paper "Boundary loss for highly unbalanced segmentation" by H. Kervadec et al. You could cite the published work in the abstract and in the extended version draw parallel or compare the distance map loss penalty to boundary loss.

---

### Official Review · AnonReviewer1 · 2019-05-01
**Distance map based loss function penalty for knee MRI segmentation**

**Rating:** 3
**Confidence:** 2

**Review:**

The paper aimed to penalize the cross-entropy loss function by distance maps to improve bone segmentation results in knee MRI. A V-Net trained and tested with the penalized loss function was compared against V-Net trained with cross-entropy, Dice loss, and focal loss functions, and showed improved results in terms of distance. Data was obtained from the keen MRI dataset of the Osteoarthritis Initiative, and the task focused on keen bone and patella segmentation.

Pros:

1. A simple-to-implement penalty term that improved network training and performance based on distance maps.

2. The paper is clear.

Cons:

1. While the results of bone segmentation are encouraging, cartilage segmentation could be a much better target. Ultimately for this work with a simple incremental idea for the loss function and the proposed title, validation in multiple challenging applications is expected. Ideally, testing on benchmark (challenge) datasets would show if the proposed method is competitive. This was not done here, but could significantly improve the confidence and interest in this work.

2. In theory the distance map penalty could be applied to other loss functions such as Dice. It would be interesting to examine this.

---

### Decision · Program_Chairs · 2019-05-06
**Acceptance Decision**

Accept